# A Biomimetic Nonwoven-Reinforced Hydrogel for Spinal Cord Injury Repair

**DOI:** 10.3390/polym14204376

**Published:** 2022-10-17

**Authors:** Ben Golland, Joanne L. Tipper, Richard M. Hall, Giuseppe Tronci, Stephen J. Russell

**Affiliations:** 1Institute of Medical and Biological Engineering, University of Leeds, Leeds LS2 9JT, UK; 2School of Biomedical Engineering, University of Technology Sydney, Sydney, NSW 2007, Australia; 3School of Mechanical Engineering, University of Leeds, Leeds LS2 9JT, UK; 4Clothworkers Centre for Textile Materials Innovation for Healthcare, School of Design, University of Leeds, Leeds LS2 9JT, UK; 5School of Dentistry, St. James’s University Hospital, University of Leeds, Leeds LS9 7TF, UK

**Keywords:** SCI, spinal cord, hydrogel, glycidylmethacrylated collagen, electrospun, nonwoven, PCL, SAP, P_11_-8

## Abstract

In clinical trials, new scaffolds for regeneration after spinal cord injury (SCI) should reflect the importance of a mechanically optimised, hydrated environment. Composite scaffolds of nonwovens, self-assembling peptides (SAPs) and hydrogels offer the ability to mimic native spinal cord tissue, promote aligned tissue regeneration and tailor mechanical properties. This work studies the effects of an aligned electrospun nonwoven of P_11_-8—enriched poly(**ε**-caprolactone) (PCL) fibres, integrated with a photo-crosslinked hydrogel of glycidylmethacrylated collagen (collagen-GMA), on neurite extension. Mechanical properties of collagen-GMA hydrogel in compression and shear were recorded, along with cell viability. Collagen-GMA hydrogels showed J-shaped stress–strain curves in compression, mimicking native spinal cord tissue. For hydrogels prepared with a 0.8-1.1 wt.% collagen-GMA concentration, strain at break values were 68 ± 1–81 ± 1% (±SE); maximum stress values were 128 ± 9–311 ± 18 kPa (±SE); and maximum force values were 1.0 ± 0.1–2.5 ± 0.1 N (±SE). These values closely mimicked the compression values for feline and porcine tissue in the literature, especially those for 0.8 wt.%. Complex shear modulus values fell in the range 345–2588 Pa, with the lower modulus hydrogels in the range optimal for neural cell survival and growth. Collagen-GMA hydrogel provided an environment for homogenous and three-dimensional cell encapsulation, and high cell viability of 84 ± 2%. Combination of the aligned PCL/P_11_-8 electrospun nonwoven and collagen-GMA hydrogel retained fibre alignment and pore structure, respectively, and promoted aligned neurite extension of PC12 cells. Thus, it is possible to conclude that scaffolds with mechanical properties that both closely mimic native spinal cord tissue and are optimal for neural cells can be produced, which also promote aligned tissue regeneration when the benefits of hydrogels and electrospun nonwovens are combined.

## 1. Introduction

Spinal cord injury (SCI) refers to any damage to the spinal cord [1] and affects around 1000 new patients in the UK annually [2], with a total UK SCI population estimated at 40,000 [3]. Whilst these figures reveal relatively few patients are forced to manage SCI compared to other chronic conditions—such as cancer, diabetes or heart disease—costs to the NHS are in excess of £1 billion [4]. This is due to the severity of injury requiring multiple and varied symptom management for the rest of a patient’s life.

The spinal cord is designed to transmit information between the brain and periphery, reflected in both the macroscale [1,5] and microscale structure of spinal cord tissue [6], and highlights the strong structure-function relationship. Disturbance to this structure from an SCI thus disrupts the normal function of information transmission between the brain and periphery and comes from the primary injury itself and the resulting biological response. Regardless of how the injury occurs, at the micronscale axons and blood vessels are damaged, and the subsequent release of damage-associated molecular patterns (DAMPs), such as cell debris, ATP and other moieties usually located inside cells, initiates a secondary biological response [7,8]. This response is a hugely complex series of events that continuously alters over time and is influenced by injury severity. This has been well detailed elsewhere [7,8,9] and its modification is the focus of some research groups through transplantation of stem cells [10] and enzymes [11], but in short represents a catastrophic scattering of cell debris, breach of the blood-spinal cord barrier, oedema, ischemia, multiple immune-mediated responses and, ultimately, development of the characteristic spinal injury scar. It is the loss of native structure from initial injury and subsequent reorganisation of cells and ECM to form the spinal injury scar that causes loss of function.

To date, only two biomaterial scaffolds have been implanted into humans to address regeneration—the Neuro-Spinal [12] and NeuroRegen scaffolds [13,14,15,16,17]. The Neuro-Spinal scaffold consists of a highly porous matrix of poly(lactic-co-glycolic acid) (PLGA) conjugated to poly(l-lysine) and achieved moderate success. In a cohort of 16 acute SCI patients, 7 saw improved American Spinal Cord Injury Association (ASIA) Impairment Scale (AIS) scores, indicating a downgrade in injury severity [12]. Similarly, the NeuroRegen scaffold, consisting of decellularised bovine collagen, implanted with human umbilical cord mesenchymal stem cells (hUCB-MSCs), achieved improved AIS scores in 11 of 20 acute cervical patients [16].

However, results for the secondary endpoints collected in the Neuro-Spinal trial were underwhelming and significant motor recovery was still regarded as unattainable. Even weaker outcomes were observed in other trials of the NeuroRegen scaffold, with no improvement in AIS score for 6 acute SCI patients implanted alongside autologous bone marrow mononuclear cells (BMMCs) [17], 5 chronic SCI patients implanted alongside BMMCs [13] and 8 chronic SCI patients implanted alongside hUCB-MSCs [14].

These results indicate that the implantation of a scaffold appears to have some positive effect, though stem cell source and injury type influence outcomes. Overall, the limited regeneration observed from the minimal clinical data may be due to the lack of a mechanically optimised environment for cell survival and tissue regeneration that has become an important research avenue of regenerative medicine [18]. When integrated with directional cues, this could be a more potent promoter of regeneration than directional cues on their own, used in the NeuroRegen scaffold [13,14,15,16,17], or a porous environment lacking both mechanical optimisation and directional cues, as in the Neuro-Spinal scaffold [12,19,20]. Addressing these emerging design requirements, biomimetic composite scaffolds comprising of both fibres and hydrogels are amongst the most promising.

Recently, the incorporation of SAPs with a degradable polymer-based electrospun nonwoven have been shown to provide beneficial properties with respect to hard tissue repair; specifically P_11_-8 peptides and PCL (PCL/P_11_-8) [21,22,23]. However, such scaffolds have only been produced and studied with randomly oriented fibres, rather than the highly aligned geometry needed for spinal cord tissue regeneration [24,25,26,27,28,29].

Similarly, collagen with glycidyl methacrylate (GMA) residues conjugated to its backbone (collagen-GMA) is able to form a photo-crosslinked hydrogel that has shown beneficial properties as a wound dressing material and as a regenerative scaffold [30,31], but has not been investigated for three-dimensional culture of neural cells or spinal cord tissue. If an aligned electrospun nonwoven of PCL/P_11_-8 could be produced and partnered with an optimised collagen-GMA hydrogel capable of three-dimensional cell culture, it was hypothesised their respective properties and integration as a scaffold would be beneficial for SCI repair.

Accordingly, the aim of this study was to develop and characterise a biomimetic scaffold comprised of functionalised collagen-GMA hydrogel and aligned PCL/P_11_-8 electrospun nonwoven fibres to recapitulate the aligned structure of spinal cord tissue in vitro in three dimensions.

## 2. Materials and Methods

### 2.1. Materials

PCL, 80,000 Mw (Sigma Aldrich, Gillingham, UK); 1,1,1,3,3,3-hexafluoro-2-propanol (HFIP), purity ≥ 99.0% (Sigma Aldrich, Gillingham, UK); P_11_-8 (CH_3_CO-Gln-Gln-Arg-Phe-Orn-Trp-Orn-Phe-Glu-Gln-Gln-NH_2_), SAP content 75%, HPLC purity of 96% (C.S. Bio Co., Menlo Park, CA, USA); Poly-D-lysine (PDL) (Sigma Aldrich, Gillingham, UK); RPMI 1640 (Lonza, Slough, UK); Horse serum, heat-deactivated (Sigma Aldrich, Gillingham, UK); Fetal bovine serum (FBS) (Sigma Aldrich, Gillingham, UK); L-Glutamine, 200 mM (Sigma Aldrich, Gillingham, UK); Penicillin-streptomycin, 5000 U mL^−1^ (Sigma Aldrich, Gillingham, UK); Recombinant Human β-NGF (Bio-Legend, San Diego, CA, USA); Neutral buffered formalin, 10% (Sigma Aldrich, Gillingham, UK); DAPI (4,6-diamidino-2-phenylindole) (Sigma Aldrich, Gillingham, UK); Triton X-100 (Sigma Aldrich, Gillingham, UK); Phosphate buffered saline (PBS) (Sigma Aldrich, Gillingham, UK); Glacial acetic acid (CH_3_COOH) (Sigma Aldrich, Gillingham, UK); GMA (Sigma Aldrich, Gillingham, UK); Triethylamine (Sigma Aldrich, Gillingham, UK); Tween-20 (Sigma Aldrich, Gillingham, UK); Ethanol (Sigma Aldrich, Gillingham, UK); 2-Hydroxy-1-[4-(2-hydroxyethoxy)phenyl]-2-methylpropan-1-one (I2959) (Fluorochem, Hadfield, UK); Dimethyl sulfoxide (DMSO) (Thermo Fisher Scientific, Altrincham, UK); Dulbecco’s modified Eagle’s medium (DMEM), High Glucose (Lonza, Slough, UK); Distilled water (Distilled in-house); Rat tail collagen type-I (Extracted in-house post-mortem); LIVE/DEAD™ Viability/Cytotoxicity Kit for mammalian cells (Invitrogen™, Waltham, MA, USA); Mouse Anti-Neuron-specific β-III Tubulin Monoclonal Antibody (clone TuJ-1), MAB1195 (Bio-Techne, Minneapolis, MN, USA); Polyclonal Rabbit Anti-Mouse Immunoglobulins/FITC, F0232 (Dako, Glostrup, DK).

### 2.2. Cells

Cell lines were supplied by the European Collection of Authenticated Cell Cultures (ECACC). Rat PC12 ECACC 92090409 neuronal cell line was used for cell viability and neurite extension studies herein.

### 2.3. Manufacture of Electrospun Nonwoven Scaffolds

Aligned electrospun nonwoven scaffolds were manufactured to serve as a directional guide for neurite extension. Aligned in this context relates to the preferential fibre orientation in the scaffold relative to the machine direction (perpendicular to the axis of the rotating drum). PCL was dissolved in HFIP overnight by stirring to form a 6% w/w spinning solution for use, or with P_11_-8 (SAP content 75%, HPLC purity of 96%) (C.S. Bio Co., Menlo Park, CA, USA) added into the spinning solution at 40 mg mL^−1^ and stirred for a further 24 h before use, in accordance with previous work [21,22,23]. The spinning solution was loaded in to a 10 mL syringe (Sigma-Aldrich, Gillingham, UK) fitted with an 18 gauge blunt needle and connected to a Model 200 Series syringe pump (KD Scientific, Holliston, MA, USA). To electrospin, the loaded syringe was connected to a power supply (Glassman High Voltage, High Bridge, NJ, USA) and placed opposite a grounded collector with the following spinning parameters; flow rate, 1 mL h^−1^; needle-to-collector distance, 110 mm; applied voltage, 25 kV.

To control the degree of fibre alignment, a rotating drum (300 mm × 200 mm, l × ø) spinning at up to 150 rad s^−1^ was used to collect aligned electrospun webs. The material was collected for 2 h, which produced a web-like sheet of nonwoven thick enough to handle and retrieved by cutting the web across the width of the collector to deliver 600 mm× 200 mm long samples for further evaluation and measurement. Sample thickness was measured using a desktop SEM to ensure similar thickness (SD ± 10%) between samples and no obvious difference in alignment on the front and back.

### 2.4. Manufacture of Collagen-GMA Hydrogel

Collagen-GMA hydrogels were manufactured to serve as a space-filling component of the nonwoven-reinforced hydrogel and allow three-dimensional encapsulation of cells. Manufacture of collagen-GMA hydrogels required extraction of type I collagen from rat tail tendons, chemical functionalisation with GMA, and solubilisation and photo-activation in a photo-initiator-supplemented aqueous solution.

Collagen type-I was isolated in-house from rat tail tendons as a readily available and inexpensive source of collagen [32]. Frozen rat tails were obtained from Central Biological Services at the University of Leeds, thawed in distilled water, skin discarded and tendons removed. Tendons were placed in 17.4 mM acetic acid (Sigma-Aldrich, Gillingham, UK) and stirred for 72 h at 4 °C to dissolve the collagen type-I. The solution was centrifuged at 38,500× *g* for 60 min, pellet discarded and supernatant lyophilised for 5 days to obtain collagen type-I. Lyophilised collagen type-I was stored at 4 °C for up to 3 months.

Collagen was functionalised with GMA to enable controlled cross-linking between glycidylmethacrylated collagen molecules. Lyophilised rat tail collagen type-I was solubilised in 17.4 mM acetic acid (0.25 weight percent (wt.%), 100 g solution) for 24 h at room temperature. Solution was neutralised dropwise to pH 7.4 using 1 M and 10 mM NaOH (Sigma-Aldrich, Gillingham, UK) at room temperature to enhance the reactivity of primary amino groups of collagen lysines. GMA (Sigma-Aldrich, Gillingham, UK) was added at a 25-molar excess with respect to collagen lysine content, as per previous work indicating further excess did not translate into increased functionalisation [31]. This was followed by an equimolar amount of triethylamine (Sigma-Aldrich, Gillingham, UK) as a non-nucleophilic base and 1 wt.% Tween-20 (Sigma-Aldrich, Gillingham, UK) with respect to initial solution mass as a surfactant to improve GMA miscibility, which have shown together to improve reaction yield [30]. After 24 h at room temperature, the solution was precipitated in a ten-fold volume excess of ethanol (Sigma-Aldrich, Gillingham, UK) and stirred for a further 24 h to remove unreacted species [30]. Solution was centrifuged at 26,500× *g* for 40 min, supernatant discarded and collagen-GMA lyophilised for 24 h. Lyophilised collagen-GMA was used immediately.

Photo-activation and the resulting cross-linking between collagen-GMA molecules was achieved using UV light and I2959 photo-initiator, as the absorption of a photon cleaves I2959 into substituent radicals. PBS was used as the solvent to mimic physiological conditions [30,31] and allow three-dimensional cell encapsulation. I2959 was dissolved in PBS prior to collagen-GMA addition as this required heating to 60 °C in a water bath and stirring for 3 h to fully dissolve. Once cooled to room temperature, lyophilised collagen-GMA (0.8–1.6 wt.%) was solubilised in I2959-PBS (Lonza, Slough, UK) (1.0 wt.%) for 24 h at room temperature. Collagen-GMA hydrogel-forming solution was used to fill silicone moulds sandwiched between glass slides and UV-irradiated (Spectroline, 365 nm, 9 mW cm^−2^) for 10 min.

### 2.5. Rheology of Collagen-GMA Hydrogel

Rheology of collagen-GMA hydrogels was studied to identify the relationship between weight percent of collagen-GMA in hydrogels and the resulting shear modulus, as well as the most appropriate weight percent for a spinal cord tissue scaffold. Hydrogels were created by UV-curing as above in silicone moulds sandwiched between glass slides to create shallow cylinders that were loaded in to an MCR302 Modular Compact Rotational Rheometer (Anton Paar, Graz, Austria). Fully hydrated hydrogels were trimmed to 25 mm diameter to match the dimensions of the 25 mm diameter parallel measuring plate. Sample height was approximately 1.5 mm and samples were compressed by 10% to ensure satisfactory grip between the plate and sample. Amplitude sweeps were run at 10 rad s^−1^ across 0.01–10% strain to identify the critical strain and preceding linear viscoelastic region (TA Instruments, 2013). Frequency sweeps were subsequently run at an amplitude within the identified linear viscoelastic region across 0.1–100 rad s^−1^ to obtain storage (*G′*) and loss (*G″*) modulus values, as within the linear viscoelastic region the storage modulus is largely independent of frequency (TA Instruments, 2013). Complex shear modulus values were calculated using the following formula:*G** = √*G*^′2^ + *G*^″2^
where *G** is the complex shear modulus, *G′* is the storage modulus and *G″* is the loss modulus. All samples were run at 37 °C and a solvent trap was filled with distilled water to prevent dehydration.

### 2.6. Compression of Collagen-GMA Hydrogel

Compression of collagen-GMA hydrogels was studied to identify the relationship between weight percent of collagen-GMA in hydrogels and the stress–strain behaviour, including the stress–strain profile, strain at break and maximum stress. Discs of hydrogel were created by UV-curing as above in silicone moulds sandwiched between glass slides. Fully hydrated hydrogels were trimmed to 5 mm diameter and compressed between two parallel plates using a Bose ElectroForce^®^ 3200, fitted with a 10 N load cell. Sample height was approximately 1.5 mm. Strain rate was 1% s^−1^. All samples were compressed to failure.

### 2.7. Cell Viability of PC12 Cells in Collagen-GMA Hydrogel

Cell viability was studied to determine whether three-dimensional cell encapsulation was possible with the collagen-GMA hydrogel system. PC12 cells were expanded in RPMI 1640 media (Lonza, Slough, UK) supplemented with 10% heat-deactivated horse serum (Sigma-Aldrich, UK), 5% FBS (Sigma-Aldrich, Gillingham, UK), 1% L-Glutamine (Sigma-Aldrich, Gillingham, UK) and 1% penicillin/streptomycin (Sigma-Aldrich, Gillingham, UK). Cells were mixed with collagen-GMA hydrogel-forming solution at 3 × 10^6^ cells mL^−1^. Collagen-GMA hydrogel-forming solution at 0.8 wt.% would normally contain 0.8 wt.% lyophilised collagen-GMA, 1.0 wt.% I2959 and 98.2% PBS. To ensure no extra volume of liquid was added when cells were mixed with collagen-GMA hydrogel-forming solution, 88.2 wt.% of PBS was added during hydrogel production and 10.0% of cell-laden media was loaded into the collagen-GMA hydrogel-forming solution and triturated thoroughly to ensure even distribution, whilst avoiding production of air bubbles. Positive controls, where a cytotoxic response was expected, were prepared as above but with 50% (*v/v*) DMSO in the cell culture media. PC12 cells grown on tissue culture polystyrene coated in PDL to aid adherence were used as negative controls, with high viability expected.

Cell-laden collagen-GMA hydrogel-forming solution was used to fill silicone moulds sandwiched between glass slides and UV-irradiated as above for 10 min. Cell-laden hydrogels were allowed to fully hydrate in pre-warmed media and transferred to an incubator at 37 °C/5% CO_2_. Cells were allowed to grow for 48 h with one 50% media change after 24 h. A LIVE/DEAD™ Viability/Cytotoxicity Kit for mammalian cells (Invitrogen™, Waltham, MA, USA) was used to stain live cells green using calcein-AM and dead cells red with ethidium homodimer-1. DAPI was used to visualize cell nuclei. Cells were imaged using a Leica SP8 Confocal Laser Scanning Microscope. ImageJ software (National Institutes of Health, Bethesda, MD, USA) was used to count the number of cells and given as a percentage of total cell number.

### 2.8. Neurite Extension of PC12 Cells in Hydrogel and Nonwoven-Reinforced Hydrogel

Neurite extension was studied as an in vitro proxy to determine whether collagen-GMA hydrogel could support spinal cord tissue regeneration. Cell-laden gels were incubated in the presence of recombinant human β-NGF (BioLegend, San Diego, CA, USA), added to the media at 100 ng mL^−1^. Media was changed every 2–3 days. After 7 days, cells were fixed by immersing in 10% neutral buffered formalin (Sigma-Aldrich, Gillingham, UK) for 15 min, and permeabilised by immersing in 0.2% Triton X-100 (Sigma-Aldrich, Gillingham, UK) for 20 min. Samples were stained using Mouse Anti-Neuron-specific β-III Tubulin Monoclonal Antibody (clone TuJ-1) (MAB1195, Bio-Techne, Minneapolis, MN, USA) for 1 h at room temperature, followed by Polyclonal Rabbit Anti-Mouse Immunoglobulins/FITC (F0232, Dako, Glostrup, Denmark) for 1 h at room temperature in the dark. Cells were imaged using a Leica SP8 Confocal Laser Scanning Microscope.

### 2.9. Manufacture of the Composite Nonwoven-Reinforced Hydrogel

An aligned electrospun nonwoven was manufactured to serve as a directional guide for neurite extension within the nonwoven-reinforced hydrogel (Figure 1). Samples of nonwoven were cut out using a 10 mm circular punch and sandwiched between silicone moulds, themselves sandwiched between glass slides. Collagen-GMA hydrogel-forming solution was then mixed with PC12 cells and UV-irradiated (Spectroline, 365 nm, 9 mW cm^−2^) for 10 min, placed in pre-warmed media and incubated at 37 °C/5% CO_2_.

For lyophilised samples, scaffolds were washed and allowed to full hydrate in distilled water for 24 h, frozen at −80 °C for 24 h and lyophilised for a further 24 h.

### 2.10. Nonwoven, Hydrogel and Nonwoven-Reinforced Hydrogel Characterisation

SEM imaging using a Hitachi SU8230 of electrospun nonwoven was employed to study fibre morphology, diameter and fibre alignment. Electrospun sample thickness was measured using a micrometer. Hydrogels and nonwoven-reinforced hydrogels were washed and allowed to fully hydrate in distilled water for 24 h, frozen at −80 °C for 24 h and lyophilised for a further 24 h. SEM imaging using a Hitachi SU8230 was employed to study pore architecture, fibre alignment and integration between the hydrogel and nonwoven.

## 3. Results

SEM images of electrospun PCL showed uniform submicron diameter fibres (Figure 2A). Similar uniform submicron diameter fibres were observed for PCL/P_11_-8, but alongside a secondary network of nanoscale diameter fibres (Figure 2B). High alignment in the machine direction was observed for submicron diameter networks for both PCL (Figure 2C) and PCL/P_11_-8 (Figure 2D), with 75% and 65% of fibres ±10° of 0°, respectively. PCL had a unimodal distribution of fibre diameters with a mean fibre diameter (±SD) of 743 ± 139 nm (Figure 2E). PCL/P_11_-8 had a bimodal distribution of fibre diameters, owing to the secondary nanoscale network, with a mean fibre diameter (± SD) of 418 ± 95 nm and 22 ± 8 nm for the respective peaks (Figure 2F). Sample thickness (± SD) was 22 ± 2 μm and 30 ± 3 μm for PCL and PCL/P_11_-8, respectively.

Scanning electron micrograph (SEM) images of lyophilised collagen-GMA hydrogel showed a sponge-like scaffold that was hundreds of microns thick, with pore structure maintained throughout (Figure 3A). Pores around 100 μm in diameter were clearly evident (Figure 3B).

Collagen-GMA hydrogels were compressed to observe the effect of weight percent on the stress–strain behaviour of the hydrogels and to compare to the literature on spinal cord tissue. Collagen-GMA hydrogels of 0.8, 0.9, 1.0 and 1.1 wt.% were compressed to failure. J-shaped stress–strain curves were observed for all four weight percent groups (Figure 4A–D). Strain at break was 68 ± 1, 75 ± 2, 71 ± 3 and 81 ± 1% (±SE), respectively. Maximum stress was 128 ± 9, 219 ± 8, 246 ± 20 and 311 ± 18 kPa (±SE), and maximum force was 1.0 ± 0.1, 1.7 ± 0.1, 2.0 ± 0.2 and 2.5 ± 0.1 N (±SE), respectively.

Rheology was also used to observe the effect of collagen-GMA concentration on hydrogel stiffness. Following identification of the linear viscoelastic region, frequency sweeps showed higher storage modulus values in relation to the loss modulus values for all concentrations, indicating a dominance of elastic strength over viscous strength and that all concentrations were behaving as viscoelastic solids (Figure 5A–F). Increasing storage and loss modulus values with increasing weight percent of collagen-GMA was observed. At an angular frequency of 0.1 rad s^−1^, collagen-GMA at 0.8, 0.9, 1.0, 1.1, 1.2 and 1.6 wt.% resulted in hydrogels with storage modulus values of 340 ± 51, 512 ± 25, 667 ± 21, 976 ± 64, 1047 ± 116 and 2574 ± 79 Pa, and loss modulus values of 61 ± 12, 56 ± 5, 79 ± 11, 88 ± 14, 97 ± 10 and 264 ± 6 Pa, respectively (Table 1). Complex shear modulus values were calculated as 345, 515, 672, 980, 1050 and 2588 Pa, respectively (Table 1).

High cell viability of 84 ± 2% was observed, as seen in the exemplary confocal images, with live cells evenly distributed in the hydrogel (Figure 6A), and throughout a 160 μm Z-stack (Figure 6B). Complete cell death was observed in the positive control and complete cell survival was observed in the negative control (not shown).

Cells were observed throughout the collagen-GMA hydrogel immediately after incubation (Figure 7A), and after 7 days could be seen extending neurites into the hydrogel (Figure 7B). Some neurites were over 250 μm in length (Figure 7C) and were observed extending in multiple planes (Figure 7D).

SEM images of the lyophilised nonwoven-reinforced hydrogel when cross-sectioned in the direction of fibre alignment show a scaffold 300–350 μm in thickness, with pores of collagen-GMA hydrogel appearing connected to both sides of the nonwoven (Figure 8A). Further, using higher magnification and focusing on the nonwoven, integration is confirmed as sheets of collagen-GMA hydrogel can be seen in between adjacent nonwoven fibres when cross-sectioned perpendicular to fibre alignment (Figure 8B).

For the first time, electrospun PCL/P_11_-8 nonwoven and collagen-GMA hydrogel were combined to form a nonwoven-reinforced hydrogel that supported directional neurite extension. Cells were observed both on the nonwoven surface (Figure 9A) and away from the nonwoven surface surrounded by hydrogel (Figure 9B). After 7 days of NGF exposure, cells in contact with the nonwoven surface were visualised extending neurites in the direction of fibre alignment (Figure 9C), whilst cells away from the nonwoven surface were observed extending neurites in a random orientation in the hydrogel (Figure 9D).

## 4. Discussion

It is possible to imagine a future scenario where a clinician orders a product for implantation in the spinal cord, as part of routine clinical practice for treatment following SCI. This product would be an off-the-shelf, pre-packaged and sterilized medical device to promote aligned tissue regeneration and thus restore function. Indeed, the Neuro-Spinal ([10], ClinicalTrials.gov NCT03762655) and NeuroRegen ([11,12,13,14,15], ClinicalTrials.gov NCT02510365) scaffolds currently in clinical trials are the first generation of such products. However, as their preclinical development began more than a decade ago, recent research themes were not incorporated as part of their design. Recent research themes include conductive hydrogels [33], hydrogels designed to improve the survival of transplanted stem cells [10], and hydrogels designed to modify the fibrotic microenvironment [11]. In contrast, this study is concerned with the mechanical optimisation of the hydrogel component, and importantly, combining with an aligned electrospun nonwoven for directional cues.

For the scaffold developed herein, mechanical properties were investigated in compression and shear. In compression, J-shaped stress–strain curves are typical of many biological tissues [34], and though limited data exists, this includes spinal cord [35,36,37,38]. The J-shape indicates that a large increase in strain is observed for a relatively short increase in stress at stresses far away from the failure point, but much larger stress is required for further increases in strain close to the failure point. The benefit of this stress–strain behaviour is highly deformable tissue during normal everyday stresses and protection against injury when stresses are extreme [39]. It has recently been reiterated that J-shaped stress–strain behaviour should be mimicked by regenerative scaffolds to produce implants that match both the everyday movement and injury-prevention characteristics of these tissues [39].

Compression of the spinal cord in the transverse plane where load-deformation or stress–strain data has been displayed can be found for animals such as cats [35] and pigs [36,37], as well as more recently in human tissue [38]. Cat spinal cords were compressed at a constant rate to a defined deformation rather than to failure, but data points at large deformations (3, 4, 4.5 and 4.75 mm in 4 different cats) relative to cord thickness (5 mm) were recorded [35]. Whilst not explicitly stated in the publication, it appears a failure point was observed, remarked as a “sudden decrease in the slope of the curve (or in the rigidity of the spinal cord)” at approximately 4.25 mm, at a force of approximately 290,000 dynes.

Converting deformation to strain and force in dynes to Newtons, this occurs at approximately 85% strain and at a force of approximately 2.9 N. Going further, we can approximate a stress value at failure of 150 kPa, as the area of the indenter is given as 19.6 mm^2^ [35]. This is remarkably similar to results realised for the hydrogels tested in this work, especially the 0.8 wt.% collagen-GMA hydrogel.

Whilst this was a single animal cord, two of the other animals loaded to 4 and 4.5 mm experienced a slight plateau before decompression, which may have been an indication of failure, and occurred at similar strain values, approximately 80 and 85%, and stress values, approximately 130 and 180 kPa (force approximately 2.6 and 3.6 N, respectively). Further, it should also be noted that very little force is recorded until a deformation of 2–2.5 mm, or 40–50% strain was achieved. Again, this is remarkably like the collagen-GMA hydrogels produced in this work, which also register little force until 35–45% strain. As such, the as-produced hydrogels appear to closely mimic the compressive mechanical properties of spinal cord in the transverse plane.

Furthermore, more recently the load-deformation data in Hung et al. (1982) [35] for feline spinal cords has been replotted as stress–strain data by Cheng et al. (2008) [40]. Clear similarity can be seen in the tissue compression data up to 40% strain replotted in Cheng et al. (2008) [40] and the hydrogel data herein. Whilst the specific strain rate used in this experiment was not given, it should be noted that strain rates used throughout Hung et al. (1982) [35] were very slow (<0.0084 s^−1^), which was similar to the 1% s^−1^ used to test collagen-GMA hydrogels herein. This further indicates that collagen-GMA hydrogels adequately mimic the mechanical properties of spinal cord tissue at low strains.

Similar J-shaped stress strain curves have also been observed when compressing porcine spinal cord in the same transverse plane [37]. Strain at break values appeared similar to those for collagen-GMA hydrogels, occurring at between 61 ± 5% and 75 ± 7% depending on vertebral level and strain rate. Similar maximum stress values were also recorded, ranging from approximately 150–500 kPa, depending on vertebral level and strain rate. The similarity of the values to those of feline samples recorded by Hung et al. (1982) [35] indicates spinal cord mechanical properties may be well preserved across mammals.

However, it should be noted that the strain rates used by Fradet et al. (2016) [37] were 50% s^−1^ at the lowest, up to 500% and 5000%. As spinal cord is known to display viscoelastic properties in compression, with both rat [41] and porcine [36,37] spinal cord becoming stiffer with increased strain rate, a direct comparison to the hydrogels tested at 1% s^−1^ herein should be made cautiously.

Similar J-shaped stress–strain curves have also been observed specifically for the white matter of porcine spinal cord [36]. Whilst not explicitly stating the plane in which samples were tested, sample height in the transverse plane was given as approximately 1.5 mm with a sample diameter of 3 mm, and as such it has been assumed testing was done at 90° to the angle of axon alignment rather than dorsal-ventral as with Hung et al. (1982) [35] and Fradet et al. (2016) [37]. Strain values of approximately 1, 2, 3 and 4 kPa up to 40% strain were recorded for strain rates of 0.5, 5, 50 and 500% s^−1^. This again highlights the importance of recording strain rate and where appropriate comparisons can be made, but also highlights the similarity of collagen-GMA hydrogels to spinal cord at a similar strain rate, even when compressed in a different plane.

The J-shaped stress–strain curves observed for collagen-GMA hydrogels are also similar to those of human spinal cord [38]. However, human spinal cord is reported to have an ultimate compressive strength of around 63 ± 5 kPa and strain at break around 27% [38]. Whilst a strain rate was not given and cord diameters were also not provided to enable calculation, load was applied at 5 mm min^−1^ which is likely similar to the 1% s^−1^ used herein, if samples had a similar diameter to that of human spinal cords recorded elsewhere [42]. As such, the collagen-GMA hydrogels described herein display both a higher ultimate compressive strength and strain at break, indicating they can be deformed to a greater extent and experience greater stress than native human spinal cord tissue before failure.

However, no analysis was given, and thus it is not clear, why the values of human spinal cord reported by Karimi et al. (2017) [38] appear so different to those reported for feline [35] and porcine [36,37] spinal cord. A compressive modulus value given by Karimi et al. (2017) [38] could also not be replicated from the stress–strain data, partly due to the omission of the specific portion of the curve from which the modulus value was derived—a common and critiqued inconsistency [40]—and as such has not been used to compare to collagen-GMA hydrogels herein.

As well as mimicking the compression properties of native spinal cord tissue, collagen-GMA hydrogels at lower concentrations are also likely more favourable for three-dimensional neural cell encapsulation in terms of shear stiffness. Hydrogels with a shear modulus of a few hundred Pascals are supportive of three-dimensional neural cell culture, irrespective of hydrogel material and cell type, evidenced by chick DRG cells in agarose gels [43,44], adult rat neural stem cells (NSCs) in alginate hydrogels [45], E18 rat cortical neurons in methylcellulose-laminin hydrogels [46] and neural stem/progenitor cells (NS/PCs) in collagen type I-hyaluronan hydrogels [47]. Whilst it is less clear if lower modulus hydrogels below 200 Pa are more supportive, with evidence for [43,44] and against [46]—which may be partially cell-type dependent—evidence that higher shear modulus hydrogels are inhibiting is stronger [43,45].

The optimal stiffness values for three-dimensional cell culture are less important, as not only can these be cell-type dependent and altered by new evidence, but the results herein evidence the specificity and control in modulus values achievable when altering the amount of collagen-GMA in the hydrogel solution. Therefore, this hydrogel system is tailorable for a given purpose. Control of hydrogel stiffness using this system is possible by varying the network architecture in a number of ways; through the type of linker applied to the collagen backbone that forms the cross-link; the number of linkers attached to the backbone and the number of cross-links they can make (the degree of functionalisation); and the amount of functionalised collagen in the hydrogel [30], as performed in the current study. This presents a variety of levels at which stiffness can be manipulated [30,31,48,49].

For the purposes of spinal cord tissue regeneration and neural cell encapsulation, GMA was chosen as the linker moiety over other previously investigated moieties, such as 4-vinylbenzyl chloride (4VBC) and methacrylic anhydride (MA) [30,31,48,49,50], due to its greater elasticity, resulting in hydrogels with greater compressibility and reduced compressive modulus [30,31]. As such, it was rationalised GMA would produce hydrogels closer to the desired shear modulus.

Further, whilst the degree of functionalisation can be controlled by varying the excess of linker used in the reaction with respect to collagen lysine content [30,31,49], it was rationalised that it would be better to reach a plateau with respect to the concentration-functionalisation relationship, as this would ensure a similar degree of functionalisation was achieved when batch producing lyophilised collagen-GMA. Indeed, it had already been shown an excess of GMA above 25 molar with respect to collagen lysine content did not result in considerable further functionalisation [30,31], and thus a 25 molar excess was used herein.

As such, with the linker moiety and molar excess taken as set parameters, this left the amount of collagen-GMA used in the hydrogel solution as the most reliable, and therefore the most appropriate, method of altering hydrogel stiffness. This was observed herein by the positive linear relationship observed between weight percent and shear modulus values.

For the first time, cells were encapsulated in collagen-GMA hydrogel. Cell viability was above the 70% recommended to indicate a non-cytotoxic effect [51]. This was despite exposing cells to UV-irradiation during hydrogel curing, to encapsulate cells throughout the hydrogel. This was achieved as stable gels were obtained after 10 min, minimising the overall dose of UV received by cells and limiting radical-induced toxicity. Dose was calculated using the following formula:*D* = *I* × *t*
where *D* is UV dosage in J cm^−2^, *I* is UV intensity in W cm^−2^ and *t* is time in seconds. Using this formula, PC12 cells encapsulated in collagen-GMA hydrogels subjected to 9 mW cm^−2^ of UV for 10 min herein received a UV dose of 5.4 J cm^−2^. This is at the lower end of the 5–10 J cm^−2^ doses typically used in cell-based photochemistry applications [52].

These results were similar to other studies using similar materials and conditions; such as BMSCs in a dual-phase hydrogel of collagen-GMA and hyaluronic acid-MA (7 mW cm^−2^, 5 min, 2.1 J cm^−2^, I2959) [53]; chondrocytes in glycidyl acrylate functionalised poly(vinyl) alcohol (8 mW cm^−2^, 10 min, 4.8 J cm^−2^, I2959) [54]; smooth muscle cells in PEGylated fibrinogen hydrogels (5–20 mW cm^−2^, 5 min, up to 1.5–6 J cm^−2^, I2959) [55]; and chondrocytes in gelatin-MA (31 mW cm^−2^, up to 12 min, up to 22.3 J cm^−2^, I2959) [56]. As such, the study herein reinforces the understanding that cells can sustain some UV exposure when used as a reaction tool and cell viability can be maintained at a high level.

Beyond cell viability though, UV-irradiation is known to induce DNA damage [57], and therefore whilst a cytotoxic dose may not have been received, genotypic changes could still have occurred that promote latent cell death, affect genetic stability, or even initiate morbidity. Recently, however, it has been shown that a UV dose at the lower end of the window used for cell-based photochemistry applications likely does not induce changes in gene expression itself, but rather it is the radical generation that results in gene expression changes, including those related to signalling, DNA damage and cell cycle [52].

This point has been exemplified recently in a study looking at cartilage regeneration using 3D printed gelatin-MA [56]. Cell viability remained high up to a dose of 5.58 J cm^−2^ (31 mW cm^−2^, 3 min) but after 4.65 J cm^−2^ (31 mW cm^−2^, 2.5 min) 50% of cells had DNA-damage, as per detection of the DNA damage marker γ-H2αX. Whilst the latter genotoxic study was performed on monolayers of cells and the former cytotoxic study on encapsulated cells, and therefore a direct comparison should be made cautiously, there is an indication cell viability can look high whilst also masking high levels of genotoxicity. Thus, whilst cell viability remained high after a UV dose of 5.4 J cm^−2^ for PC12 cells encapsulated in collagen-GMA hydrogel herein, DNA damage may well have occurred.

However, similar to the fact some UV exposure is tolerable to cells before becoming cytotoxic, there is an indication cells can be subjected to some level of UV dose without any measurable DNA damage as well. For example, after a UV dose of 1.86 J cm^−2^ no γ-H2αX was measured in chondrocytes, and this could well be higher as the next experimental group measured was not until a UV dose of 4.65 J cm^−2^ [56]. In relation to the collagen-GMA hydrogels herein, this indicates further optimisation below 10 min UV-irradiation time, which has not yet been investigated, may result in fully cross-linked hydrogels whilst minimising genotoxicity.

Further, this is without even considering other factors that could influence toxicity; such as reducing the amount of photoinitiator [54,58]; the fact different cell types respond differently to radical-induced toxicity [58]; that depending on scaffold dimensions and manufacturing method prolonged UV exposure may be required [56] or that other photoinitiator systems also exist [59]. Indeed, the photoinitiator lithium phenyl-2,4,6-trimethylbenzoylphosphinate (LAP) has been shown to cure diacrylated poly(ethylene glycol) hydrogels faster at a similar UV wavelength (365 nm) and also cure within the visible spectrum (405 nm), further reducing the likelihood of toxicity [60,61]. LAP has also been shown to cure gelatin methacrylol (GelMa) hydrogels whilst supporting neural cells [62]. However, it should be noted that LAP at the time of writing is around 20 times the cost of I2959 and may be an influencing factor on photo-initiator system choice.

However, despite the potential genotoxic effects, PC12 cells were observed extending long neurites in multiple planes, showcasing the three-dimensional cell culture potential of this hydrogel system.

SEM images of the nonwoven-reinforced hydrogel showed fibres had become wavy, as tension had been lost after punching out the samples, but alignment appeared to remain intact and resembled the high alignment in the machine direction observed after electrospinning on to the rotating drum. Importantly, other studies which had used layering to combine nonwovens with hydrogels and saw some loss of tension, or the degree of alignment had been less than in the current study, did not appear to hinder process extension from neurons or other glial cells [63,64] and was similar to those studies where tension and alignment had been maintained [65,66]. As such, integration of collagen-GMA hydrogel with PCL/P_11_-8 fibres was confirmed and fibre alignment was not disturbed.

Further, integration led to a scaffold with improved usability at the macroscale, as handling of the nonwoven-reinforced hydrogel versus the hydrogel alone was much easier. This is an early indication that clinical handling of the scaffold would be improved along with conformability to the tissue following implantation on to the defect.

The spatial effect of layering hydrogel and nonwoven on manipulating neurite extension agrees with the results observed elsewhere. Neurite alignment from SH-SY5Y neuron-like cells was influenced greater in a scaffold of electrospun PCL and hyaluronic acid hydrogel when cells were in close contact with fibres, and augmented when fibres were coated in laminin [63]. Similarly, neurite elongation from rat cortical neurons was observed in a composite scaffold of aligned chitosan hydrogel layered over aligned electrospun PLLA, the effects of layering most notable when each material was layered perpendicular to each other and neurites were observed extending in the direction of alignment relative to the focal plane [66]. This effect has also been observed with glial cells, where rat astrocytes aligned relative to the direction of aligned fibres in a stacked composite scaffold of electrospun poly-L,D-lactic acid and collagen hydrogel, and complex process-bearing oligodendrocytes were also observed contacting aligned fibres and astrocytes [65].

Whilst the effects were not quantified herein, there was an indication that cells did not have to be in direct contact with fibres to be directionally influenced. This has been somewhat quantified in a different study, where a composite scaffold of aligned PCL fibres with a PuraMatrix™ SAP-based hydrogel layered on top observed human embryonic stem cell (hESC)-derived neurons aligning neurites with the direction of the underlying fibres when up to 10 μm away but no further [64].

Additionally, regarding fibre orientation, the appearance of the secondary network of nanoscale fibres observed for PCL/P_11_-8 that was not aligned in the machine direction did not appear to affect neurite extension in the direction of submicron fibre alignment. The secondary network has been reported previously for randomly orientated webs but never for aligned webs prior to the work herein [19,20,21].

As such, the results herein show a nonwoven-reinforced hydrogel of electrospun PCL/P_11_-8 fibres and collagen-GMA hydrogel are able to support directed neurite outgrowth.

However, whilst layering can produce a nonwoven-reinforced hydrogel capable of influencing cell morphology and process alignment from cells, it should be highlighted these effects are regionally defined. Indeed, this has been labelled as a critique of layering as an approach for combining nonwoven and hydrogel, as materials are still limited to surface or near-surface effects rather than a more homogeneous three-dimensional environment at the cell scale [67]. The results herein reinforce the understanding that layering, especially at the scale performed herein, can influence cell and process alignment from cells, but does not provide a homogeneous environment in which all cells are exposed to directional cues.

However, the author challenges the notion this is a major drawback to layering as an approach for creating a nonwoven-reinforced hydrogel, especially for spinal cord injury repair. The spinal cord itself is made up of regions where promoting alignment would be beneficial for recapitulating the tissue structure, such as in the white matter, but also of regions where alignment is only partially required, such as in the grey matter. As such, defining regions within a scaffold where alignment would be necessary and beneficial, but also defining those regions where alignment would possibly promote abnormal tissue structuring compared to the native environment, seems just as important. As such, whilst not taken further in the work herein, the author would suggest that regional variation in the ability of a scaffold to promote neurite extension be looked at as an advantage for controlling tissue structuring, especially for tissues such as spinal cord, where heterogeneity is apparent and important for the structure-function relationship.

## 5. Conclusions

It is possible to imagine a future where an off-the-shelf SCI regeneration product is included in routine clinical practice. However, products currently undergoing clinical trials do not incorporate recent design trends, including the combination of fibres and hydrogels to mimic biological tissue, and a mechanically optimised environment. The nonwoven-reinforced hydrogel of PCL/P_11_-8 and collagen-GMA developed herein demonstrated retained pore structure of the hydrogel and retained fibre alignment of the electrospun nonwoven when the two were combined. This resulted in homogenous and three-dimensional cell encapsulation, high cell viability, and aligned neurite extension. Mechanical properties in compression mimicked the J-shaped nature of native tissue, as well as strain-at-break and maximum strain values of animal tissue. Mechanical properties in shear could be tailored to fall in the range optimal for neural cell survival and growth. Application of such nonwoven-reinforced hydrogels to other tissues may also be applicable.

## Figures and Tables

**Figure 1 polymers-14-04376-f001:**
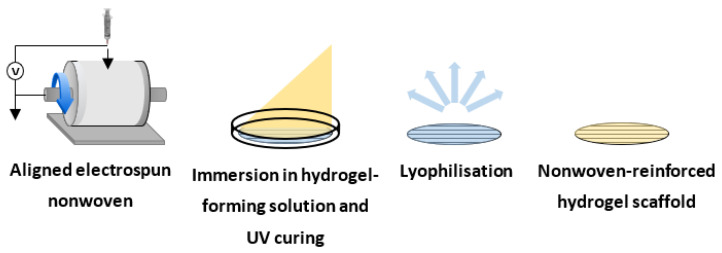
Diagram showing the scaffold manufacturing process. Nonwoven is electrospun on to a rotating drum to highly align fibres. The sheet of highly aligned nonwoven is then immersed in hydrogel-forming solution using silicone moulds sandwiched between glass slides and UV cured. Lastly, the scaffold is lyophilized to produce the nonwoven-reinforced hydrogel scaffold.

**Figure 2 polymers-14-04376-f002:**
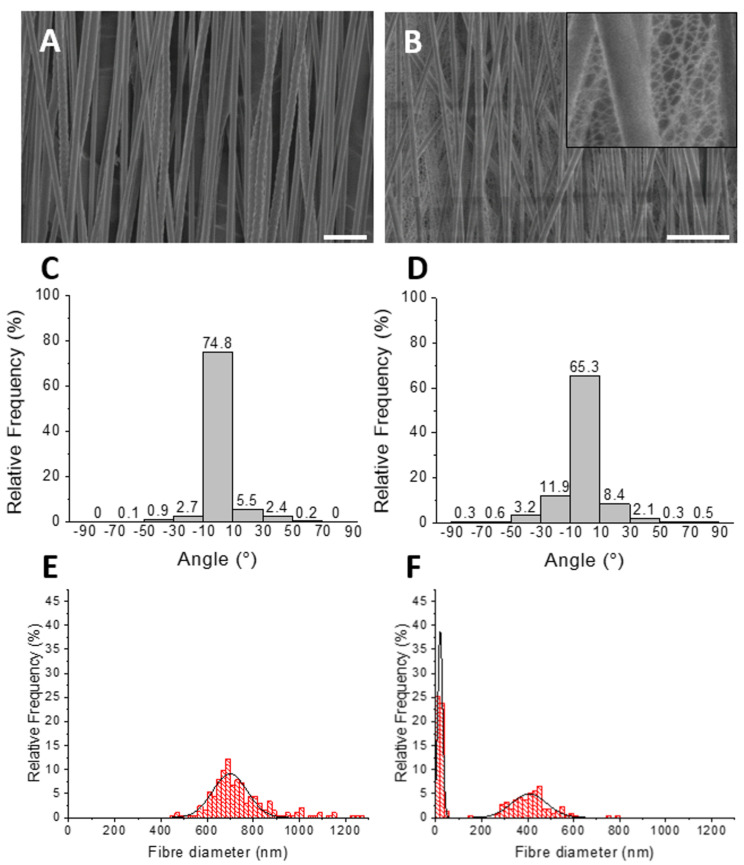
SEM images, fibre alignment and fibre diameter distribution of electrospun PCL and PCL/P_11_-8 nonwovens. High alignment of the submicron fibre network in the machine direction can be seen in both (**A**) PCL and (**B**) PCL/P_11_-8 40 mg mL^−1^. The appearance of a secondary nanoscale network can be seen throughout the PCL/P_11_-8 web (inset). Scale bar is 5 µm. Alignment of the primary submicron fibre network for (**C**) PCL and (**D**) PCL/P_11_-8 40 mg mL^−1^ was 75% and 65% ± 10° of 0°, respectively. *n* = 100. (**E**) PCL had a unimodal distribution of fibre diameters versus a bimodal distribution of fibre diameters for (**F**) PCL/P_11_-8 webs. *n* = 100.

**Figure 3 polymers-14-04376-f003:**
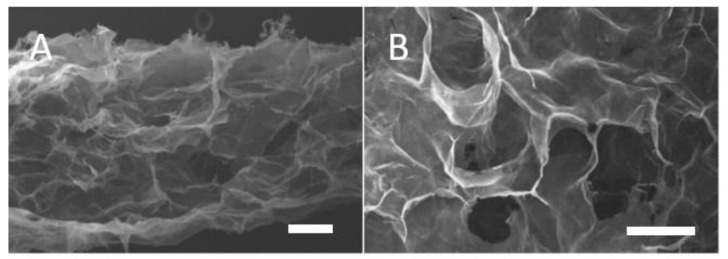
SEM images of lyophilised collagen-GMA hydrogel: (**A**) Lyophilisation formed a sponge-like collagen-GMA hydrogel that was hundreds of microns thick; (**B**) Pore structure was around 100 μm in diameter. Scale bar is 100 μm.

**Figure 4 polymers-14-04376-f004:**
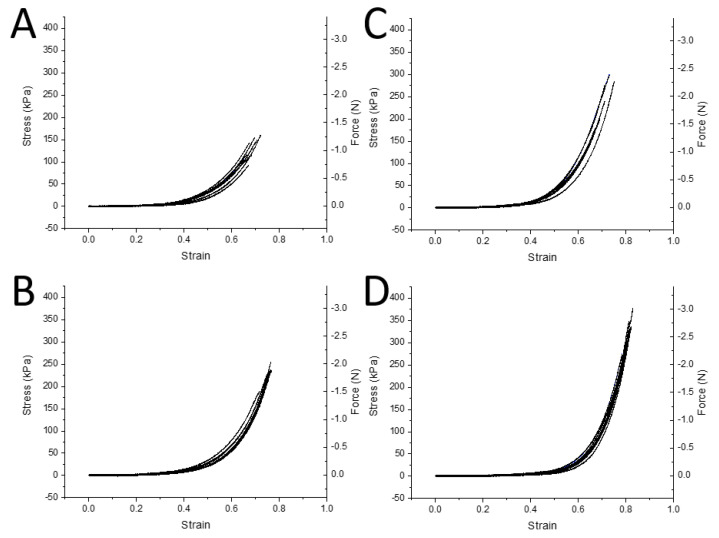
Mechanical properties in compression of collagen-GMA hydrogel at varying weight percent (wt.%). Stress-strain graphs of collagen-GMA hydrogel at (**A**) 0.8, (**B**) 0.9, (**C**) 1.0 and (**D**) 1.1 wt.%. Force has been added as a second *Y*-axis. *n* = 6–8.

**Figure 5 polymers-14-04376-f005:**
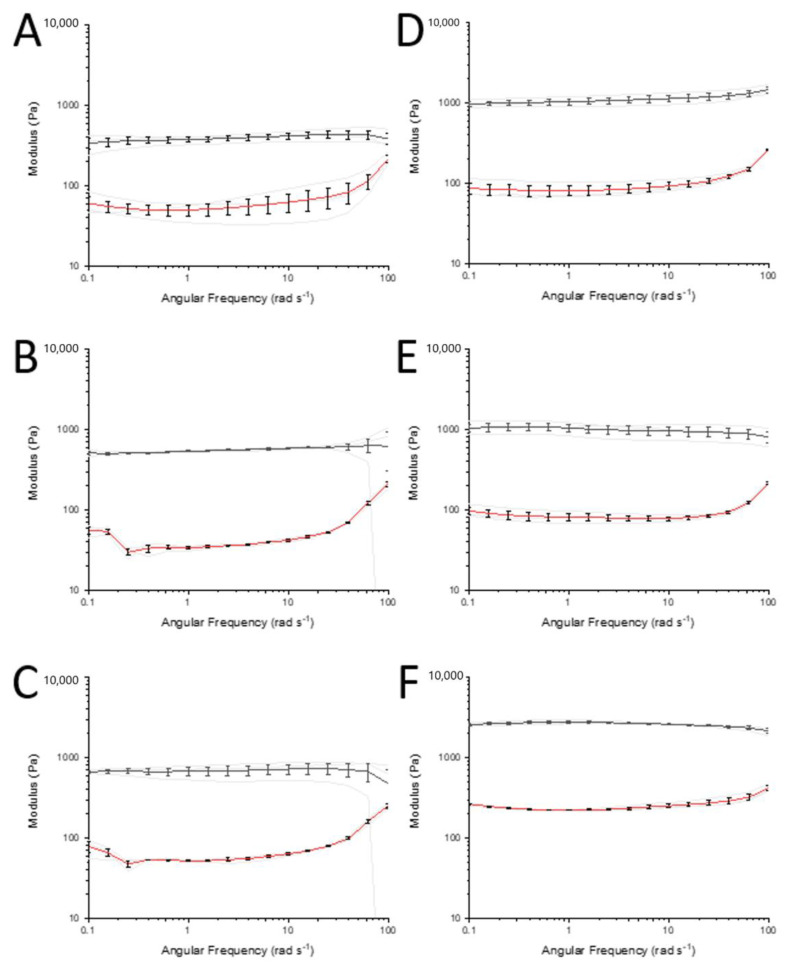
Mechanical properties in shear of collagen-GMA hydrogel at varying weight percent (wt.%). Frequency sweeps show collagen-GMA hydrogel at (**A**) 0.8, (**B**) 0.9, (**C**) 1.0, (**D**) 1.1, (**E**) 1.2 and (**F**) 1.6 wt.%. The black and red lines show the mean storage and loss modulus for each experimental group (±SE), respectively. Grey lines depict actual measurements. All samples had a higher storage modulus compared to loss modulus, indicating hydrogels were behaving as viscoelastic solids. As weight percent of collagen-GMA hydrogels increased, storage modulus also increased. *n* = 3.

**Figure 6 polymers-14-04376-f006:**
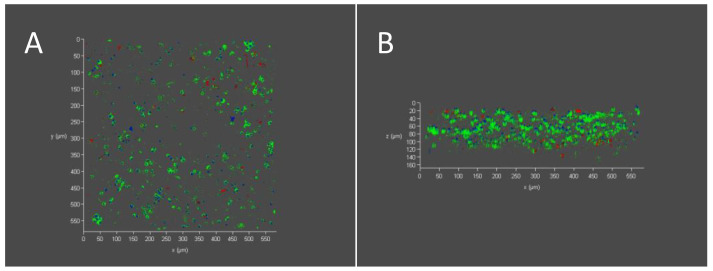
PC12 cell viability and distribution in collagen-GMA hydrogel. Exemplary confocal images show live (green) and dead (red) PC12 cells 48 h after cell seeding in 0.8 wt.% collagen-GMA hydrogel (cell nuclei stained blue). High cell viability of 84 ± 2% was observed. (**A**) Viable cells can be seen evenly distributed in the 600 µm × 600 µm section when cells were mixed with collagen-GMA hydrogel-forming solution prior to UV-irradiation. (**B**) Z-stack sections show PC12 cells throughout the 160 µm section.

**Figure 7 polymers-14-04376-f007:**
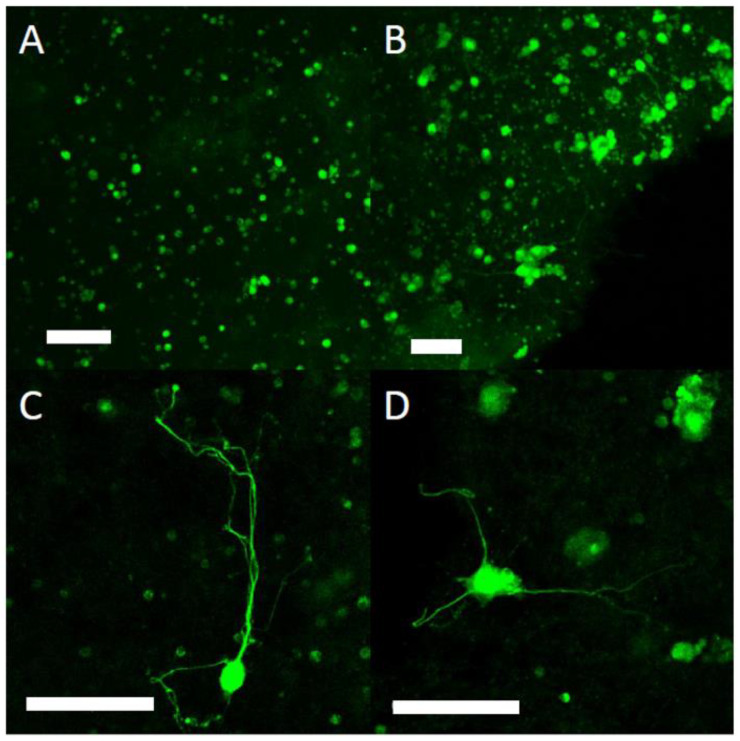
Neurite extension of PC12 cells in collagen-GMA hydrogel. PC12 cells were mixed with collagen-GMA hydrogel-forming solution, UV-irradiated for 10 min and placed into an incubator at 37 °C/5% CO_2_ for 7 days. (**A**) Immediately after incubation cells were observed throughout the collagen-GMA hydrogel. (**B**) Following 7 days of NGF exposure, PC12 cells were observed extending neurites throughout the collagen-GMA hydrogel, (**C**) with some neurites observed over 250 µm in length and (**D**) extending in multiple planes. Scale bar is 100 µm.

**Figure 8 polymers-14-04376-f008:**
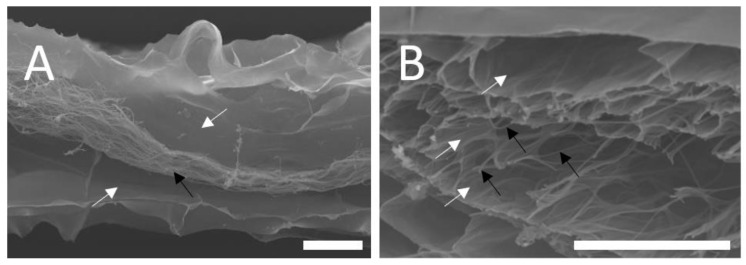
SEM images of nonwoven-reinforced hydrogel. Nonwoven-reinforced hydrogels were produced by sandwiching (**A**) PCL/P_11_-8 or (**B**) PCL electrospun nonwoven between silicone moulds and glass slides, UV-curing collagen-GMA hydrogel on both sides and lyophilising the scaffold. (**A**) Pore structures of collagen-GMA could be seen on both sides of the scaffold (white arrows), whilst alignment of the nonwoven fibres was retained (black arrow). (**B**) Higher magnification showed sheets of collagen-GMA (white arrows) between aligned nonwoven fibres (black arrows) indicating integration at the fibre level of both materials. Scale bar is 100 µm.

**Figure 9 polymers-14-04376-f009:**
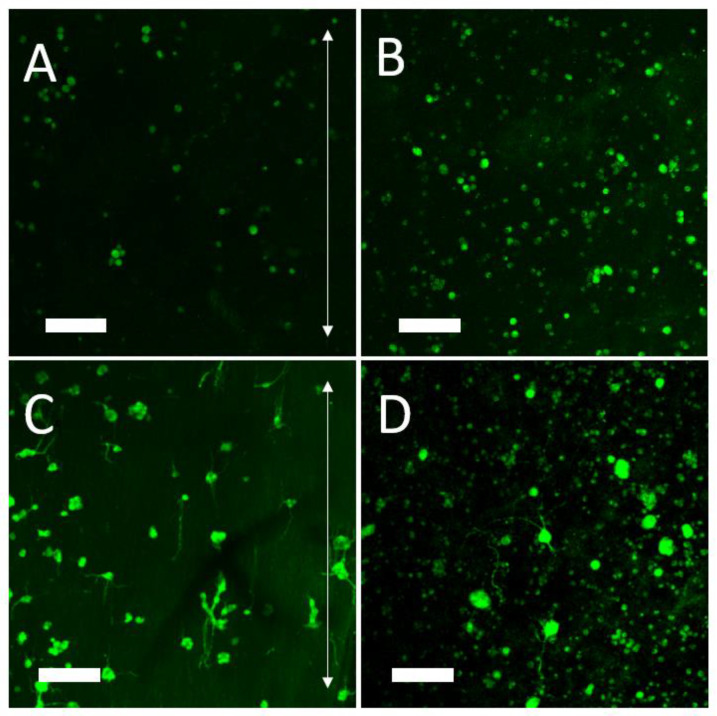
Neurite extension of PC12 cells in the composite electrospun nonwoven-hydrogel scaffold. PC12 cells were mixed with collagen-GMA hydrogel-forming solution, pipetted on top of electrospun PCL/P_11_-8 nonwoven, UV-irradiated for 10 min and placed into an incubator at 37 °C/5% CO_2_ for 7 days. Immediately after incubation cells were observed on both the (**A**) surface of electrospun PCL/P_11_-8 and (**B**) at least 50 µm away from the nonwoven surrounded by collagen-GMA hydrogel. Following 7 days of NGF exposure, (**C**) PC12 cells in contact with the nonwoven surface were observed extending neurite in the direction of fibre alignment, whilst (**D**) away from the nonwoven surface, cells were observed extending neurites in a random orientation. Arrows show direction of alignment. Scale bar is 100 µm.

**Table 1 polymers-14-04376-t001:** Table of storage, loss and complex shear modulus values of collagen-GMA hydrogel at varying weight percent (wt.%). All values as at 0.1 rad s^−1^. *n* = 3.

Collagen-GMA (wt.%)	0.8	0.9	1.0	1.1	1.2	1.6
Storage modulus ± SE (Pa)	340 ± 51	512 ± 25	667 ± 21	976 ± 64	1047 ± 116	2574 ± 79
Loss modulus ± SE (Pa)	61 ± 12	56 ± 5	79 ± 11	88 ± 14	97 ± 10	264 ± 6
Complex modulus (Pa)	345	515	672	980	1051	2588

## Data Availability

Data are available from corresponding authors upon request.

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
