# Peer review of "A Biomimetic Nonwoven-Reinforced Hydrogel for Spinal Cord Injury Repair"

_polymers, 2022, doi:10.3390/polym14204376_

Round 1

Reviewer 1 Report

From the article, I saw that spinal cord repair could be achieved by regulating the mechanical properties of hydrogels, but there was no difference in cell culture caused by different ratio of hydrogels. 

The paper lacks the support of animal experimental results and cannot confirm the effect of the described advantages in actual restoration.

1. This article has certain innovation. The biomimetic non-woven enhanced hydrogel is combined with photoinitiated hydrogel and electrospinning to optimize the properties and functions of hydrogel.

2. The logical organization and language use of the article should be improved.

3. The paper lacks sufficient data to support its so-called optimized hydrogel properties, and there are no further animal models to prove its function in repairing nerve cells.

4. The references chosen for the article can be considered appropriatly.

Author Response

Dear Ms. Haily Wang,   

Thank you very much for your feedback on our manuscript (Manuscript ID: polymers-1932246) entitled “A biomimetic nonwoven-reinforced hydrogel for spinal cord injury repair”, and for the important suggestions kindly provided by the Reviewers, which have helped us to improve the quality of the work. We have revised the manuscript to address Reviewers' comments and we are now submitting a revised version of the manuscript.   

Our point-by-point answers are provided below.  We hope that the revised version of the manuscript fulfils the Journal requirements of Polymers

Q1. From the article, I saw that spinal cord repair could be achieved by regulating the mechanical properties of hydrogels, but there was no difference in cell culture caused by different ratio of hydrogels.

Answer: We thank the reviewer for this query. Different hydrogel ratios were not tested in cell culture, as the literature suggests a reasonable consensus for optimal shear stiffness of a few hundred Pascals for neural cell viability (pg 14, para 4). As such, rheology was used first to identify a hydrogel ratio that had a shear stiffness in the target Pascal range, and an appropriate hydrogel ratio was thus selected for subsequent cell experiments. It was beyond the scope of the work to question the optimal shear stiffness range for neural cell viability given previously published work on this topic. We make the point in the discussion that optimal shear stiffness is likely cell-type dependent (pg 14, para 5), and therefore the real property of value for a hydrogel system is one that can be easily tuned to a different stiffness, as displayed in the manuscript (Fig. 5). The cell culture experiments presented were specifically conducted to show cell viability (Fig. 6), cell distribution (Fig. 6) and neurite extension (Fig. 7) of neuron-like cells in a hydrogel that had not been used for this purpose before. It was also shown that when the hydrogel was combined with an aligned electrospun nonwoven, cells in close proximity to the nonwoven surface extended neurites in the direction of fibre alignment, versus cells that extended neurites in a random orientation situated further away from the nonwoven surface (Fig. 9).

Q2. The paper lacks the support of animal experimental results and cannot confirm the effect of the described advantages in actual restoration.

Answer: We thank the reviewer for this query. The in vitro cell experiments were conducted to show the cell tolerability of the hydrogel system and composite nonwoven-reinforced hydrogel. Animal studies were not pursued at this stage because in this initial study, we wanted first to verify the feasibility of building a composite scaffold and to characterise its behaviour (and lack of cytotoxic effects) under lab conditions. This was to ensure its structure and properties were as intended before moving to animal studies in a next step. This systematic approach was adopted to ensure animals were not used unnecessarily, in line with the 3Rs principles (Replacement, Reduction and Refinement) of animal testing.

Q3. This article has certain innovation. The biomimetic non-woven enhanced hydrogel is combined with photoinitiated hydrogel and electrospinning to optimize the properties and functions of hydrogel.

Answer: We thank the referee for this supportive feedback. It is in light of the risks associated with the novelty of this design, that animal testing was not conducted in this study, as already discussed in the point above..

Q4. The logical organization and language use of the article should be improved.

Answer: We thank the reviewer for this query. We have now revised the manuscript and placed the original supplementary Figure S1 as Figure 2, improving the logical organisation. Furthermore, nonwoven characterisation is presented first, followed by hydrogel characterisation, and finally, characterisation of the nonwoven-reinforced hydrogel.

Q5. The paper lacks sufficient data to support its so-called optimized hydrogel properties, and there are no further animal models to prove its function in repairing nerve cells.

Answer: We thank the reviewer for this query. Mechanical optimisation as detailed in the manuscript was demonstrated by the rheological measurements, with respect to shear modulus taken from the consensus in the literature (pg 14, para 4). The manuscript demonstrates that hydrogels prepared from mixtures supplemented with 0.8 wt% collagen fall within this suggested optimal of a few hundred Pascals for neural cell viability and growth. On a more mechanistic level, Figure 5 shows the effect of collagen concentration on the shear moduli, showing the tailorability of this hydrogel system (as discussed in pg 14, para 5).

The manuscript also demonstrates that it is possible to achieve optimal shear stiffness (as described above) at the same time as achieving compressive mechanical properties that closely mimic that of native spinal cord tissue in vivo.

With respect to animal models, as answered above, this was an initial study to verify the feasibility of the design and animal studies were not pursued at this early stage to ensure animals were not used unnecessarily.

Q4. The references chosen for the article can be considered appropriatly.

Answer: We thank the referee for this comment.

Reviewer 2 Report

In this article it was aimed to produce a nonwoven aligned fiber integrated collagen-GMA hydrogel based biomaterial for spinal cord injury. The study plan is well designed and results associated to this study plan presented in an appropriate way. Some points to be addressed by the authors are as follows:

-        Applied voltage is 25V or 25kV (line 140)?

-        The dimensions of the rotating drum have not been provided.

-        How is the rotating drum’s velocity calculated as 30 m/s. It’s better to provide a RPM based rotation velocity? 30 m/s sounds very high.

-        Thick enough does not sound scientific so that it is better to provide an average with ±SD calculated using thickness measurements. Although it has been told that thickness was measured with a desktop SEM, no image of the samples or thickness measurements were provided. (Supplementary data are not reachable at the link provided- 404 error)

-     SEM image(s) should have been provided to show that the fibres are aligned as proposed. (Supplementary data are not reachable at the link provided- 404 error). A SEM image is better be presented in the article rather than supplementary files.

Author Response

Dear Ms. Haily Wang,   

Thank you very much for your feedback on our manuscript (Manuscript ID: polymers-1932246) entitled “A biomimetic nonwoven-reinforced hydrogel for spinal cord injury repair”, and for the important suggestions kindly provided by the Reviewers, which have helped us to improve the quality of the work. We have revised the manuscript to address Reviewers' comments and we are now submitting a revised version of the manuscript.   

Our point-by-point answers are provided below.  We hope that the revised version of the manuscript fulfils the Journal requirements of Polymers

Q1.  Applied voltage is 25V or 25kV (line 140)?

Answer: We thank the reviewer for pointing out the typo – voltage should be kV not V. This has now been addressed in the revised version of the manuscript (pg 3, para 4).

Q2.  The dimensions of the rotating drum have not been provided.

Answer: We thank the reviewer for pointing this out. Drum dimensions have been added to the manuscript (300 (l) × 200 (⌀) mm) (pg 3, para 5).

Q3.   How is the rotating drum’s velocity calculated as 30 m/s. It’s better to provide a RPM based rotation velocity? 30 m/s sounds very high.

Answer: We thank the reviewer for this query. Angular velocity has been represented instead as 150 rad/s (pg 3, para 5). For the avoidance of doubt, this is very high.

Q4. Thick enough does not sound scientific so that it is better to provide an average with ±SD calculated using thickness measurements. Although it has been told that thickness was measured with a desktop SEM, no image of the samples or thickness measurements were provided. (Supplementary data are not reachable at the link provided- 404 error)

Answer: We thank the reviewer for this point. Sample thickness was 22±2 μm and 30±3 μm for samples made of PCL and PCL/P11-8, respectively, and has been added to the manuscript (pg 6, para 5). These thickness measurements were taken with a micrometer (pg 6, para 4). A desktop SEM was used to simply check sample quality (alignment, thickness, uniformity) during manufacture.

Q5. SEM image(s) should have been provided to show that the fibres are aligned as proposed. (Supplementary data are not reachable at the link provided- 404 error). A SEM image is better be presented in the article rather than supplementary files.

Answer: We thank the reviewer for raising this point. The fibre images have been moved to the main text as recommended (Fig. S1 converted to Fig. 2).

Reviewer 3 Report

The paper submitted by Golland et al. deals with the preparation and investigation of a collagen hydrogel reinforced with aligned nonwoven fibers based on PCL and peptides with potential use for the repair of spinal cord injury.

The manuscript is very interesting, clear, well written and the conclusions are supported by the results. Moreover, the discussion section is very well detailed and therefore the paper can be accepted as it is.

1. The paper submitted by Golland et al. deals with the preparation of new composite scaffolds, based on peptides and PCL electrospun nonwoven fibers immobilized into a photo-crosslinked hydrogel of modified collagen, which can be used for the regeneration of spinal cord.

2. The proposed strategy is an original one and the used analyses were judiciously chosen. The obtained results demonstrate that this system promote the tissue regeneration, which was the aim of the study.

3. The paper is clear, well written and the conclusions are supported by the results.

4. Some new references could be added.

Author Response

Dear Ms. Haily Wang,   

Thank you very much for your feedback on our manuscript (Manuscript ID: polymers-1932246) entitled “A biomimetic nonwoven-reinforced hydrogel for spinal cord injury repair”, and for the important suggestions kindly provided by the Reviewers, which have helped us to improve the quality of the work. We have revised the manuscript to address Reviewers' comments and we are now submitting a revised version of the manuscript.   

Our point-by-point answers are provided below.  We hope that the revised version of the manuscript fulfils the Journal requirements of Polymers

Q1. The paper submitted by Golland et al. deals with the preparation of new composite scaffolds, based on peptides and PCL electrospun nonwoven fibers immobilized into a photo-crosslinked hydrogel of modified collagen, which can be used for the regeneration of spinal cord.

Answer: Thank you.

Q2. The proposed strategy is an original one and the used analyses were judiciously chosen. The obtained results demonstrate that this system promote the tissue regeneration, which was the aim of the study.

Answer: Thank you for highlighting the originality of the work, it is much appreciated.

Q3. The paper is clear, well written and the conclusions are supported by the results.

Answer: We thank the referee for this feedback and supportive comments. We note that this supportive feedback is in stark contrast with the comments provided by Reviewer 4 (see below).

Q4. Some new references could be added.

Answer: We thank the reviewer for this point. Yang et al. (2022), Li et al. (2022) and Park et al. (2022) have been added to the manuscript (pg 2 para 1; pg 10 para 4).

Reviewer 4 Report

The manuscript (polymers-1932246) presents the development of a biomimetic nonwoven-reinforced hydrogel for spinal cord injury repair. The developed materials were evaluated for various physicochemical and biological properties. However, the study falters at providing meaningful conclusions based on results several necessary experimentations are missing. To improve the quality of the manuscript, authors must need to address the following comments and queries:

1.     The rationale behind the study and novelty of the work must be described in detail.

2.     Structure of the manuscript leads to confusion. Manuscript structure should consist of a proper flow in terms of methodology and experimental results.

3.     Why have the authors not performed the mechanical and/or chemical characterization for nonwoven and nonwoven reinforced hydrogel? For a clear presentation and understanding of the obtained results; these comparative results should be included.

4.     Controls are missing in the in vitro studies.

5. Provided results do not well support the conclusion.

Author Response

Dear Ms. Haily Wang,   

Thank you very much for your feedback on our manuscript (Manuscript ID: polymers-1932246) entitled “A biomimetic nonwoven-reinforced hydrogel for spinal cord injury repair”, and for the important suggestions kindly provided by the Reviewers, which have helped us to improve the quality of the work. We have revised the manuscript to address Reviewers' comments and we are now submitting a revised version of the manuscript.   

Our point-by-point answers are provided below.  We hope that the revised version of the manuscript fulfils the Journal requirements of Polymers

Q1. The rationale behind the study and novelty of the work must be described in detail.

Answer: The rationale for the work is set out in the introduction (line 59-96), detailing how (a) current scaffolds in the clinic have not led to significant functional recovery; (b) these scaffolds do not incorporate leading research themes and this may be the reason for the aforementioned results; (c) highly complex structures are required to meet the complex design requirements and have not been well explored; and (d) recent work on separate electrospun nonwoven and hydrogel systems may hold promise to address these needs, but their integration, optimisation and characterisation for spinal cord regeneration had not been explored at all – the aim and novelty of the manuscript (line 97-100). For further detail:

  • Current scaffolds in the clinic have not led to significant functional recovery. Briefly, only two biomaterial scaffolds have been implanted into humans to address regeneration to date and results have been underwhelming for both.
  • These scaffolds do not incorporate leading research themes. Whilst, the importance of a mechanically optimised environment has emerged as a leading research theme for regenerative medicine and, likewise, directional cues have been shown to direct neurite outgrowth, the scaffolds in the clinic do not incorporate both these design requirements. Neither of the scaffolds in the clinic have been mechanically characterised as per the published data and only one has directional cues. To address these complex design requirements – incorporation of both mechanical optimisation and directional cues – composite scaffolds comprising both fibres and hydrogels are amongst the most promising.
  • These are highly complex structures and integration of materials capable of achieving these requirements have not yet been well explored. It appeared that the recent incorporation of self-assembling peptides (SAPs) into a PCL-based electrospun nonwoven may be able to provide adequate directional cues, and a collagen-based hydrogel functionalised with glycidyl methacrylate may be able to provide adequate mechanical properties. However, these systems had yet to be integrated, or optimised and characterised for spinal cord tissue regeneration – separately or together.

Further, the manuscript also shows it is possible to achieve a hydrogel with a shear stiffness optimal for neural cell viability (as per the consensus in the literature, pg 14 para 4) at the same time as mechanical properties in compression (J-shaped stress-strain curve, similar strain at break, similar maximum stress, similar onset of stress response) that match native spinal cord tissue (in vivo experiments from the literature). This was unexpected, surprising and novel.

Q2. Structure of the manuscript leads to confusion. Manuscript structure should consist of a proper flow in terms of methodology and experimental results.

Answer: We thank the reviewer for this query. The methodology is reported, followed by the experimental results section as is the normal convention. Optimisation and characterisation of the two separate components (electrospun nonwoven and hydrogel) were conducted first before combining and characterising as the composite nonwoven-reinforced hydrogel. The manuscript originally placed the nonwoven characterisation as Supplementary Materials (Fig. S1), but to further ensure a logical flow and structure, this has now been moved to the main text (Fig. 2).

For the avoidance of doubt, the overall nonwoven-reinforced hydrogel manufacturing process is first presented – Fig. 1; followed by characterisation of the nonwoven – Fig. 2; followed by characterisation of the hydrogel (SEM images of the collagen hydrogel – Fig. 3; mechanical properties in shear – Fig. 4; mechanical properties in compression – Fig. 5; cell viability and distribution – Fig. 6; neurite extension – Fig. 7); followed by characterisation of the nonwoven-reinforced hydrogel (SEM images showing integration of the nonwoven and hydrogel – Fig. 8; neurite extension close to the nonwoven surface and further away – Fig. 9).

Q3.  Why have the authors not performed the mechanical and/or chemical characterization for nonwoven and nonwoven reinforced hydrogel? For a clear presentation and understanding of the obtained results; these comparative results should be included.

Answer: We thank the reviewer for this query. For a spinal cord regenerative scaffold, the mechanical properties of the component being used to space-fill the cavity and surround cells is of most importance – the hydrogel in this case. As per the literature, there is a consensus around optimal shear stiffness for neural cell viability (a few hundred Pascals, pg 14 para 4) and spinal cord tissue mechanical properties in compression have been investigated in vivo. The inclusion of a thin sheet of electrospun nonwoven in the scaffold would have little effect on the bulk mechanical properties of a nonwoven-reinforced hydrogel as presented, and the effect at the cell-scale would also be minimal. The electrospun nonwoven fibres have been incorporated as directional cues and to enhance the clinical handling and conformability to the tissue following implantation on to the defect (pg 16 para 4). 

Q4. Controls are missing in the in vitro studies.

Answer: We thank the reviewer for this query. Information regarding controls for cell viability (Fig. 6) are included in the Results (pg 9, para 2). Figure 7 simply shows neurite extension in the hydrogel for exemplary purposes. Similarly, Figure 9 shows the effect of neurite orientation when grown close to, and further away from the nonwoven surface for exemplary purposes. 

Q5. Provided results do not well support the conclusion.

Answer: We thank the reviewer for this query. The conclusion states as per the results:

  • The alignment of the nonwoven (Fig. 2) and the pore structure of the hydrogel (Fig. 3) are both retained when integrated together to form the nonwoven-reinforced hydrogel (Fig. 8).
  • Homogenous and three-dimensional cell encapsulation (Fig. 6), high cell viability (Fig. 6), and aligned neurite extension in the hydrogel (Fig. 7) and nonwoven-reinforced hydrogel (Fig. 9).
  • Mechanical properties in compression (Fig. 4) mimicked the J-shaped nature of native tissue, as well as strain-at-break and maximum strain values of animal tissue as per the literature.
  • Mechanical properties in shear (Fig. 5) could be tailored to fall in the range optimal for neural cell survival and growth as per the literature.

Round 2

Reviewer 1 Report

I am fully satisfied by the revision

Reviewer 4 Report

The manuscript is significantly improved. The authors have addressed all the queries raised.